# A NON-ASYMPTOTIC COMPARISON OF SVRG AND SGD: TRADEOFFS BETWEEN COMPUTE AND SPEED

## ABSTRACT

Stochastic gradient descent (SGD), which trades off noisy gradient updates for computational efficiency, is the de-facto optimization algorithm to solve large-scale machine learning problems. SGD can make rapid learning progress by performing updates using subsampled training data, but the noisy updates also lead to slow asymptotic convergence. Several variance reduction algorithms, such as SVRG, introduce control variates to obtain a lower variance gradient estimate and faster convergence. Despite their appealing asymptotic guarantees, SVRG-like algorithms have not been widely adopted in deep learning. The traditional asymptotic analysis in stochastic optimization provides limited insight into training deep learning models under a fixed number of epochs. In this paper, we present a non-asymptotic analysis of SVRG under a noisy least squares regression problem. Our primary focus is to compare the exact loss of SVRG to that of SGD at each iteration $t$. We show that the learning dynamics of our regression model closely matches with that of neural networks on MNIST and CIFAR-10 for both the underparameterized and the overparameterized models. Our analysis and experimental results suggest there is a trade-off between the computational cost and the convergence speed in underparametrized neural networks. SVRG outperforms SGD after the first few epochs in this regime. However, SGD is shown to always outperform SVRG in the overparameterized regime.

## 1 INTRODUCTION

Many large-scale machine learning problems, especially in deep learning, are formulated as minimizing the sum of loss functions on millions of training examples (Krizhevsky et al., 2012; Devlin et al., 2018). Computing exact gradient over the entire training set is intractable for these problems. Instead of using full batch gradients, the variants of stochastic gradient descent (SGD) (Robbins & Monro, 1951; Zhang, 2004; Bottou, 2010; Sutskever et al., 2013; Duchi et al., 2011; Kingma & Ba, 2014) evaluate noisy gradient estimates from small mini-batches of randomly sampled training points at each iteration. The mini-batch size is often independent of the training set size, which allows SGD to immediately adapt the model parameters before going through the entire training set. Despite its simplicity, SGD works very well, even in the non-convex non-smooth deep learning problems (He et al., 2016; Vaswani et al., 2017). However, the optimization performance of the stochastic algorithm near local optima is significantly limited by the mini-batch sampling noise, controlled by the learning rate and the mini-batch size. The sampling variance and the slow convergence of SGD have been studied extensively in the past (Chen et al., 2016; Li et al., 2017; Toulis & Airoldi, 2017). To ensure convergence, machine learning practitioners have to either increase the mini-batch size or decrease the learning rate toward the end of the training (Smith et al., 2017; Ge et al., 2019).

Recently, several clever variance reduction methods (Roux et al., 2012; Defazio et al., 2014; Wang et al., 2013; Johnson & Zhang, 2013) were proposed to alleviate the noisy gradient problem by using control-variates to achieve unbiased and lower-variance gradient estimators. In particular, the variants of Stochastic Variance Reduced Gradient (SVRG) (Johnson & Zhang, 2013), k-SVRG (Raj & Stich, 2018), L-SVRG (Kovalev et al., 2019) and Free-SVRG (Sebbouh et al., 2019) construct control-variates from previous staled snapshot model parameters. These methods enjoy a superior asymptotic performance in convex optimization compared to the standard SGD. The control-variate techniques are shown to improve the convergence rate of SGD from a sub-linear to a linear convergence rate. These variance reduction methods can also be combined with momentum (Allen-Zhu, 2017) and preconditioning methods (Moritz et al., 2016) to obtain faster convergence. Despite

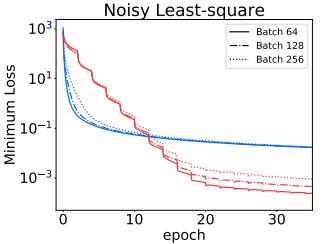 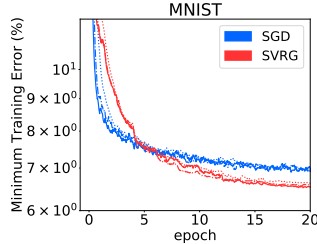

(a) Least-squares regression (predicted).    (b) Logistic regression.

Figure 1: (a) The minimum loss achieved over a set of hyperparameters in a noisy least squares regression problem (simulated dynamics). (b) The minimum loss achieved in real dataset MNIST (a logistic regression model). Our theoretical prediction (a) matched with the training dynamics for real datasets, demonstrating tradeoffs between computational cost and convergence speed. The curves in red are SVRG and curves in blue are SGD. Different markers refer to different per-iteration computational cost, i.e., the number of backpropagation used per iteration on average.

their strong theoretical guarantees, SVRG-like algorithms have seen limited success in training deep learning models (Defazio & Bottou, 2018). Traditional results from stochastic optimization focus on the asymptotic analysis, but in practice, most of deep neural networks are only trained for hundreds of epochs due to the high computational cost. To address the gap between the asymptotic benefit of SVRG and the practical computational budget of training deep learning models, we provide a non-asymptotic study on the SVRG algorithms under a noisy least squares regression model. Although optimizing least squares regression is a basic problem, it has been shown to characterize the learning dynamics of many realistic deep learning models (Zhang et al., 2019; Lee et al., 2019). Recent works suggest that neural network learning behaves very differently in the underparameterized regime vs the overparameterized regime (Ma et al., 2018; Vaswani et al., 2019), characterized by whether the learnt model can achieve zero expected loss. We account for both training regimes in the analysis by assuming a linear target function and noisy labels. In the presence of label noise, the loss is lower bounded by the label variance. In the absence of the noise, the linear predictor can fit each training example perfectly. We summarize the main contributions as follows:

- We show the exact expected loss of SVRG and SGD along an optimization trajectory as a function of iterations and computational cost.

- Our non-asymptotic analysis provides an insightful comparison of SGD and SVRG by considering their computational cost and learning rate schedule. We discuss the trade-offs between the total computational cost, i.e. the total number of back-propagations performed, and convergence performance.

- We consider two different training regimes with and without label noise. Under noisy labels, the analysis suggests SGD only outperforms SVRG under a mild total computational cost. However, SGD always exhibits a faster convergence compared to SVRG when there is no label noise.

- Numerical experiments validate our theoretical predictions on both MNIST and CIFAR-10 using various neural network architectures. In particular, we found the comparison of the convergence speed of SGD to that of SVRG in underparameterized neural networks closely matches with our noisy least squares model prediction. Whereas, the effect of overparameterization is captured by the regression model without label noise.

## 1.1 RELATED WORKS

**Stochastic variance reduction methods** consider minimizing a finite-sum of a collection of functions using SGD. In case we use SGD to minimize these objective functions, the stochasticity comes from the randomness in sampling a function in each optimization step. Due to the induced noise, SGD can only converge using decaying step sizes with sub-linear convergence rate. Methods such as SAG (Roux et al., 2012), SVRG (Johnson & Zhang, 2013), and SAGA (Defazio et al., 2014), are able to recover linear convergence rate of full-batch gradient descent with the asymptotic cost comparable to SGD. SAG and SAGA achieve this improvement at the substantial cost of storing the most recent gradient of each individual function. In contrast, SVRG spends extra computation at snapshot intervals by evaluating the full-batch gradient. Theoretical results such as Gazagnadou et al. (2019) show that under certain smoothness conditions, we can use larger step sizes with stochastic variance reduction methods than is allowed for SGD and hence achieve even faster convergence. In situations where we know the smoothness constant of functions, there are results on the optimal mini-batch size and the optimal step size given the inner loop size (Sebbouh et al., 2019). Applying variance

reduction methods in deep learning has been studied recently (Defazio & Bottou, 2018). The authors conjectured the ineffectiveness is caused by various elements commonly used in deep learning such as data augmentation, batch normalization and dropout. Such elements can potentially decrease the smoothness and make the stored gradients become stale quickly. The proposed solution is to either remove these elements or update the gradients more frequently than is practical.

**Dynamics of SGD and quadratic models** Our main analysis tool is very closely related to recent work studying the dynamics of gradient-based stochastic methods. Wu et al. (2018) derived the dynamics of stochastic gradient descent with momentum on a noisy quadratic model (Schaul et al., 2013), showing the problem of short horizon bias. In (Zhang et al., 2019), the authors showed the same noisy quadratic model captures many of the essential characteristic of realistic neural networks training. Their noisy quadratic model successfully predicts the effectiveness of momentum, preconditioning and learning rate choices in training ResNets and Transformers. However, these previous quadratic models assume a constant variance in the gradient that is independent of the current parameters and the loss function. It makes them inadequate for analyzing the stochastic variance reduction methods, as SVRG can trivially achieve zero variance under the constant gradient noise. Instead, we adopted a noisy least-squares regression formulation by considering both the mini-batch sampling noise and the label noise. There are also recent works that derived the risk of SGD, for least-squares regression models using the bias-variance decomposition of the risk (Belkin et al., 2018; Hastie et al., 2019). We use a similar decomposition in our analysis. In contrast to the asymptotic analysis in these works, we compare SGD to SVRG along the optimization trajectory for any finite-time horizon under limited computation cost, not just the convergence points of those algorithms.

**Underparameterization vs overparameterization.** Many of the state-of-the-art deep learning models are overparameterized deep neural networks with more parameters than the number of training examples. Even though these models are able to overfit to the data, when trained using SGD, they generalize well (Zhang et al., 2017). As suggested in recent work, underparameterized and overparameterized regimes have different behaviours (Ma et al., 2018; Vaswani et al., 2019; Schmidt & Roux, 2013). Given the infinite width and a proper weight initialization, the learning dynamics of a neural network can be well-approximated by a linear model via the neural tangent kernel (NTK) (Jacot et al., 2018; Chizat & Bach, 2018). In NTK regime, neural networks are known to achieve global convergence by memorizing every training example. On the other hand, previous convergence results for SVRG have been obtained in stochastic convex optimization problems that are similar to that of an underparameterized model (Roux et al., 2012; Johnson & Zhang, 2013). Our proposed noisy least-squares regression analysis captures both the underparameterization and overparameterization behavior by considering the presence or the absence of the label noise.

## 2 PRELIMINARY

### 2.1 NOTATIONS

We will primarily focus on comparing the minibatch version of two methods, SGD and SVRG (Johnson & Zhang, 2013). Denote $L_i$ as the loss on $i^{th}$ data point. The SGD update is written as,

$$\boldsymbol{\theta}^{(t+1)} = \boldsymbol{\theta}^{(t)} - \alpha^{(t)}\hat{\mathbf{g}}^{(t)}, \tag{1}$$

where $\hat{\mathbf{g}}^{(t)} = \frac{1}{b}\sum_i^b \nabla_{\boldsymbol{\theta}^{(t)}} L_i$ is the minibatch gradient, $t$ is the training iteration, and $\alpha^{(t)}$ is the learning rate. The SVRG algorithm is an inner-outer loop algorithm proposed to reduce the variance of the gradient caused by the minibatch sampling. In the outer loop, for every $T$ steps, we evaluate a large batch gradient $\bar{\mathbf{g}} = \frac{1}{N}\sum_i^N \nabla_{\boldsymbol{\theta}^{(mT)}} L_i$, where $N \gg b$, and $m$ is the outer loop index, and we store the parameters $\boldsymbol{\theta}^{(mT)}$. In the inner loop, the update rule of the parameters is given by,

$$\boldsymbol{\theta}^{(mT+t+1)} = \boldsymbol{\theta}^{(mT+t)} - \alpha^{(t)}\left(\hat{\mathbf{g}}^{(mT+t)} - \tilde{\mathbf{g}}^{(mT+t)} + \bar{\mathbf{g}}\right) \tag{2}$$

where $\hat{\mathbf{g}}^{(mT+t)} = \frac{1}{b}\sum_i^b \nabla_{\boldsymbol{\theta}^{(mT+t)}} L_i$ is the current gradient of the mini-batch and $\tilde{\mathbf{g}}^{(mT+t)} = \frac{1}{b}\sum_i^b \nabla_{\boldsymbol{\theta}^{(mT)}} L_i$ is the old gradient. Note that in our analysis, the reference point is chosen to be the last iterate of previous outer loop $\boldsymbol{\theta}^{(mT)}$, recommended as a practical implementation of the algorithm by the original SVRG paper Johnson & Zhang (2013).

## 2.2 THE NOISY LEAST SQUARES REGRESSION MODEL

We now define the noisy least squares regression model (Schaul et al., 2013; Wu et al., 2018). In this setting, the input data is $d$-dimensional, and the output label is generated by a linear teacher model with additive noise,

$$(\boldsymbol{x}_i, \epsilon_i) \sim P_x \times P_\epsilon; \quad y_i = \boldsymbol{x}_i^\top \boldsymbol{\theta}^* + \epsilon_i,$$

where $\mathbb{E}[\boldsymbol{x}_i] = \boldsymbol{\mu} \in \mathbb{R}^d$ and $\mathrm{Cov}(\boldsymbol{x}_i) = \Sigma$, $\mathbb{E}[\epsilon_i] = 0$, $\mathrm{Var}(\epsilon_i) = \sigma_y^2$. We assume WLOG $\boldsymbol{\theta}^* = \boldsymbol{0}$. We also assume the data covariance matrix $\Sigma$ is diagonal. This is an assumption adopted in many previous analysis and it is also a practical assumption as we often apply whitening to pre-process the training data. We would like to train a student model $\boldsymbol{\theta}$ that minimizes the squared loss over the data distribution:

$$\min_{\boldsymbol{\theta}} L(\boldsymbol{\theta}) := \mathbb{E}\left[\frac{1}{2}(\boldsymbol{x}_i^\top \boldsymbol{\theta} - y_i)^2\right]. \tag{3}$$

At each iteration, the optimizer can query an arbitrary number of data points $\{\boldsymbol{x}_i, y_i\}_i$ sampled from data distribution. The SGD method uses $b$ data points to form a minibatch gradient:

$$\hat{\mathbf{g}}^{(t)} = \frac{1}{b}\sum_i^b (\boldsymbol{x}_i\boldsymbol{x}_i^\top\boldsymbol{\theta}^{(t)} - \boldsymbol{x}_i\epsilon_i) = X_bX_b^\top\boldsymbol{\theta}^{(t)} - \frac{1}{\sqrt{b}}X_b\boldsymbol{\epsilon}_b, \tag{4}$$

where $X_b = \frac{1}{\sqrt{b}}[\boldsymbol{x}_1; \boldsymbol{x}_2; \cdots; \boldsymbol{x}_b] \in \mathbb{R}^{d\times b}$, and the noise vector $\boldsymbol{\epsilon}_b = [\epsilon_1; \epsilon_2; \cdots; \epsilon_b]^\top \in \mathbb{R}^b$. SVRG on the other hand, queries for $N$ data points every $T$ steps to form a large batch gradient $\bar{\mathbf{g}} = X_NX_N^\top\boldsymbol{\theta}^{(mT)} - \frac{1}{\sqrt{N}}X_N\boldsymbol{\epsilon}_N$, where $X_N$ and $\boldsymbol{\epsilon}_N$ are defined similarly. At each inner loop step, it further queries for another $b$ data points, to form the update in Eq. 2.

Lastly, note that the expected loss can be written as a function of the second moment of the iterate,

$$L(\boldsymbol{\theta}^{(t)}) = \frac{1}{2}\mathbb{E}\left[\left(\boldsymbol{x}_i^\top\boldsymbol{\theta}^{(t)} - \epsilon_i\right)^2\right] = \frac{1}{2}\left(tr(\Sigma\mathbb{E}[\boldsymbol{\theta}^{(t)}\boldsymbol{\theta}^{(t)\top}]) + \sigma_y^2\right).$$

Hence for the following analysis we mainly focus on deriving the dynamics of the second moment $\mathbb{E}[\boldsymbol{\theta}^{(t)}\boldsymbol{\theta}^{(t)\top}]$, denoted as $A(\boldsymbol{\theta}^{(t)})$. When $\Sigma$ is diagonal, the loss can further be reduced to $\frac{1}{2}\mathrm{diag}(\Sigma)^\top\mathrm{diag}(\mathbb{E}[\boldsymbol{\theta}^{(t)}\boldsymbol{\theta}^{(t)\top}]) + \frac{1}{2}\sigma_y^2$. We denote $\mathrm{diag}(\mathbb{E}[\boldsymbol{\theta}^{(t)}\boldsymbol{\theta}^{(t)\top}])$ by $\mathbf{m}(\boldsymbol{\theta}^{(t)})$.

## 2.3 THE DYNAMICS OF SGD

**Definition 1** (Formula for dynamics). *We define the following functions and identities,*

$$\mathrm{M}(\boldsymbol{\theta}) = \mathbb{E}[\boldsymbol{\theta}\boldsymbol{\theta}^\top], \quad \mathbf{m}(\boldsymbol{\theta}) = diag(\mathbb{E}[\boldsymbol{\theta}\boldsymbol{\theta}^\top]), \quad \mathrm{C}(\mathrm{M}(\boldsymbol{\theta})) = \mathbb{E}_{\boldsymbol{x}}[\boldsymbol{x}\boldsymbol{x}^\top\mathrm{M}(\boldsymbol{\theta})\boldsymbol{x}\boldsymbol{x}^\top] - \Sigma\mathrm{M}(\boldsymbol{\theta})\Sigma,$$

$$\boldsymbol{n} = \alpha^2\sigma_y^2 diag(\Sigma), \quad R = (I - \alpha\Sigma)^2 + \frac{\alpha^2}{b}(\Sigma^2 + diag(\Sigma)diag(\Sigma)^\top),$$

$$Q = \frac{2\alpha^2}{b}(\Sigma^2 + diag(\Sigma)diag(\Sigma)^\top), \quad P = I - \alpha\Sigma, \quad F = \frac{\alpha^2(N+b)}{Nb}(\Sigma^2 + diag(\Sigma)diag(\Sigma)^\top)$$

$$G = \alpha^2(\frac{b+1}{b}\Sigma^2 + \frac{1}{b}diag(\Sigma)diag(\Sigma)^\top).$$

The SGD update (Eq. 1) with the mini-batch gradient of of the noisy least squares model (Eq. 4) is,

$$\boldsymbol{\theta}^{(t+1)} = (I - \alpha X_bX_b^\top)\boldsymbol{\theta}^{(t)} + \frac{\alpha}{\sqrt{b}}X_b\boldsymbol{\epsilon}_b.$$

We substitute the update rule to derive the following dynamics for the second moment of the iterate:

$$\mathrm{M}(\boldsymbol{\theta}^{(t+1)}) = \underbrace{(I - \alpha\Sigma)\mathrm{M}(\boldsymbol{\theta}^{(t)})(I - \alpha\Sigma)}_{\text{①: gradient descent shrinkage}} + \underbrace{\frac{\alpha^2}{b}\mathrm{C}(\mathrm{M}(\boldsymbol{\theta}^{(t)}))}_{\text{②: input noise}} + \underbrace{\frac{\alpha^2\sigma_y^2}{b}\Sigma}_{\text{③: label noise}} \tag{5}$$

This dynamics equation can be understood intuitively as follows. The term ① leads to an exponential shrinkage of the loss due to the gradient descent update. Since we are using a noisy gradient, the second term ② represents the variance of stochastic gradient caused by the random input $X_b$. The

term ③ comes from the label noise. We show in the next theorem that when the second moment of the iterate approaches zero, ② will also approach zero. However due to the presence of the label noise, the expected loss is lower bounded by ③.

When $\Sigma$ is diagonal, we further analyze and decompose $C(M(\boldsymbol{\theta}))$ as a function of $\mathbf{m}(\boldsymbol{\theta})$ so as to derive the following dynamics and decay rate for SGD.

**Theorem 2** (SGD Dynamics and Decay Rate). *Given the noisy linear regression objective function (Eq. 3), under the assumption that $x \sim \mathcal{N}(\mathbf{0}, \Sigma)$ with $\Sigma$ diagonal and $\boldsymbol{\theta}^* = 0$, we can express $C(\boldsymbol{\theta})$ as a function of $\mathbf{m}(\boldsymbol{\theta})$:*

$$diag\Big(C\Big(M(\boldsymbol{\theta})\Big)\Big) = \Big(\Sigma^2 + diag(\Sigma)diag(\Sigma)^\top\Big)\mathbf{m}(\boldsymbol{\theta}) \tag{6}$$

*Then we derive following dynamics of expected second moment of $\boldsymbol{\theta}$:*

$$\mathbf{m}(\boldsymbol{\theta}^{(t)}) = R^t\Big(\mathbf{m}(\boldsymbol{\theta}^{(0)}) - \frac{(I-R)^{-1}\boldsymbol{n}}{b}\Big) + \frac{(I-R)^{-1}\boldsymbol{n}}{b}, \tag{7}$$

Under the update rule of SGD, $R$ is the decay rate of the second moment of parameters between two iterations. And based on Theorem 2 the expected loss can be calculated by $\frac{1}{2}diag(\Sigma)^\top\mathbf{m}(\boldsymbol{\theta}^{(t)}) + \frac{1}{2}\sigma_y^2$.

## 3 A DILEMMA FOR SVRG

By querying a large batch of datapoints $X_N$ every $T$ steps, and a small minibatch $X_b$ at every step, the SVRG method forms the following update rule:

$$\boldsymbol{\theta}^{(mT+t+1)} = \big(I - \alpha X_b X_b^\top\big)\boldsymbol{\theta}^{(mT+t)} + \alpha\big(X_b X_b^\top - X_N X_N^\top\big)\boldsymbol{\theta}^{(mT)} + \frac{\alpha}{\sqrt{N}}X_N\boldsymbol{\epsilon}_N \tag{8}$$

To derive the dynamics of the second moment of the parameters following the SVRG update, we look at the dynamics of one round of inner loop updates, i.e., from $\boldsymbol{\theta}^{(mT)}$ to $\boldsymbol{\theta}^{((m+1)T)}$:

**Lemma 3.** *The dynamics of the second moment of the iterate following SVRG update rule is given by,*

$$M(\boldsymbol{\theta}^{(mT+t+1)}) = \underbrace{(I-\alpha\Sigma)M(\boldsymbol{\theta}^{(mT+t)})(I-\alpha\Sigma)}_{①\ gradient\ descent\ shrinkage} + \underbrace{\frac{\alpha^2}{b}C\Big(M(\boldsymbol{\theta}^{(mT+t)})\Big)}_{②\ input\ noise} + \underbrace{\frac{\alpha^2\sigma_y^2}{N}\Sigma}_{③\ label\ noise} \tag{9}$$

$$+ \underbrace{\alpha^2\frac{N+b}{Nb}C\Big(M(\boldsymbol{\theta}^{(mT)})\Big)}_{④\ variance\ due\ to\ \tilde{\mathbf{g}}^{(mT+t)}} \underbrace{-\frac{\alpha^2}{b}\Big(C\Big(M(\boldsymbol{\theta}^{(mT)})P^t\Big) + C\Big(P^t M(\boldsymbol{\theta}^{(mT)})\Big)\Big)}_{⑤\ Variance\ reduction\ from\ control\ variate}$$

The dynamics equation above is very illuminating as it explicitly manifests the weakness of SVRG. First notice that terms ①, ②, ③ reappear, contributed by the SGD update. The additional terms, ④ and ⑤, are due to the control variate. Observe that the variance reduction term ⑤ decays exponentially throughout the inner loop, with decay rate $I - \alpha\Sigma$, i.e. $P$. We immediately notice that this is the same term that governs the decay rate of the term ①, and hence resulting in a conflict between the two. Specifically, if we want to reduce the term ① as fast as possible, we would prefer a small decay rate and a large learning rate, i.e. $\alpha \to \frac{1}{\lambda_{\max}(\Sigma)}$. But this will also make the boosts provided by the control variate diminish rapidly, leading to a poor variance reduction. The term ④ makes things even worse as it will maintain as a constant throughout the inner loop, contributing to an extra variance on top of the variance from standard SGD. On the other hand, if one chooses a small learning rate for the variance reduction to take effect, this inevitably will make the decay rate for term ① smaller, resulting in a slower convergence. Nevertheless, a good news for SVRG is that the label noise (term ③) is scaled by $\frac{b}{N}$, which lets SVRG converge to a lower loss value than SGD – a strict advantage of SVRG compared to SGD.

To summarize, the variance reduction from SVRG comes at a price of slower gradient descent shrinkage. In contrast, SVRG is able to converge to a lower loss value. This motivates the question, which algorithm to use given a certain computational cost? We hence performed a thorough investigation through numerical simulation as well as experiments on real datasets in Sec. 4.

Similarly done for SGD, we decompose $C(\boldsymbol{\theta})$ as a function of $\mathbf{m}(\boldsymbol{\theta})$ and derive the following decay rate for SVRG.

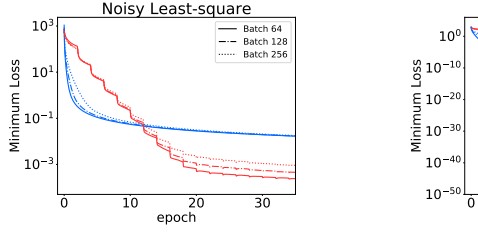 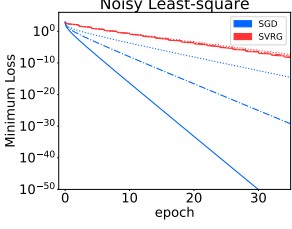

(a) With Label Noise  (b) Without Label Noise

Figure 2: The minimum loss achieved by following SGD (blue) and SVRG (red) over a set of hyperparameters in a noisy least-square dynamics simulation for cases with and without label noise. The plot suggests that in the presence of label noise, there is a tradeoff between computational cost and convergence speed. In the absence of label noise, SGD strictly dominates SVRG in convergence speed for all computational cost.

**Theorem 4** (SVRG Dynamics and Decay rate). *Given the noisy linear regression objective function (Eq. 3), under the assumption that $x \sim \mathcal{N}(\mathbf{0}, \Sigma)$ with $\Sigma$ diagonal and $\boldsymbol{\theta}^* = 0$, the dynamics for SVRG in $\mathbf{m}(\boldsymbol{\theta})$ is given by:*

$$\mathbf{m}(\boldsymbol{\theta}^{((m+1)T)}) = \lambda(\alpha, b, T, N, \Sigma)\mathbf{m}(\boldsymbol{\theta}^{(mT)}) + (I - R^T)(I - R)^{-1}\frac{\boldsymbol{n}}{N}, \tag{10}$$

$$\lambda(\alpha, b, T, N, \Sigma) = R^T - \Big(\sum_{k=0}^{T-1} R^k Q P^{-k}\Big) P^{T-1} + (I - R^T)(I - R)^{-1}F. \tag{11}$$

### 3.1 The dynamics without label Noise

In the absence of the label noise (i.e., $\sigma_y = 0$), we observe that both SGD and SVRG enjoy linear convergence as a corollary of Theorem 2 and Theorem 4:

**Corollary 5.** *Without the label noise, the dynamics of the second moment following SGD is given by,*

$$\mathbf{m}(\boldsymbol{\theta}^{(t)}) = R^t \mathbf{m}(\boldsymbol{\theta}^{(0)}),$$

*and the dynamics of SVRG is given by,*

$$\mathbf{m}(\boldsymbol{\theta}^{((m+1)T)}) = \lambda(\alpha, b, T, N, \Sigma)\mathbf{m}(\boldsymbol{\theta}^{(mT)}),$$

*where $\lambda$ is defined in Eq.( 11).*

Note that similar results have been shown in the past (Ma et al., 2018; Vaswani et al., 2019; Schmidt & Roux, 2013), where a general condition known as "interpolation regime" is used to show linear convergence of SGD. Specifically they assume that $\nabla L_i(\boldsymbol{\theta}^*) = 0$ for all $i$, and our setting without label noise clearly also belongs to this regime. This setting also has practical implications, as one can treat training overparameterized neural networks as in interpolation regime. This motivates the investigation of the convergence rate of SGD and SVRG without label noise, and was also extensively studied in the experiments detailed as follows.

## 4 Experiments

In Sec. 3 we discussed a critical dilemma for SVRG that is facing a choice between effective variance reduction and faster gradient descent shrinkage. At the same time, it enjoys a strict advantage over SGD as it converges to a lower loss. We define the total computational cost as the total number of back-propagations performed. Similarly, per-iteration computational cost refers to the number of back-propagations performed per iteration. In this section, we study the question, which algorithm converges faster given certain total computational cost? We study this question for both the underparameterized and the overparameterized regimes.

Our investigation consists of two parts. First, numerical simulations of the theoretical convergence rates (Sec. 4.1). Second, experiments on real datasets (Sec. 4.2). In both parts, we first fix the per-iteration computational cost. For SGD, the per-iteration computational budge is equal to the minibatch size. We picked three batch size $\{64, 128, 256\}$. Denote the batchsize of SGD as $b$, the equivalent batch size for SVRG is $b' = \frac{1}{2}(1 - \frac{N}{Tb})b$. We then perform training with an extensive set of hyperparameters for each method with each per-iteration computational cost. For SGD, the hyperparameter under consideration is the learning rate $\alpha$. For SVRG, besides the learning rate,

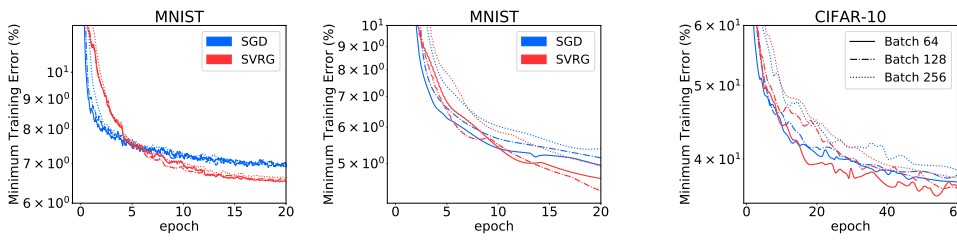

(a) Logistic Regression.      (b) underparameterized MLP      (c) underparameterized CNN

Figure 3: The minimum loss achieved by following SGD (blue) and SVRG (red) over a set of hyperparameters for training on MNIST and CIFAR-10 with underparameterized models. All the results in these plot suggested there is a tradeoff between computational cost and convergence speed when comparing SGD and SVRG.

we also ran over a set of snapshot intervals $T$. After running over all sets of hyperparameters, we gather all training curves of all hyperparameters. We then summarize the performance for each algorithm by plotting the lower bound of all training curves, i.e. each point $(l, t)$ on the curve showed the minimum loss $l$ at time step $t$ over all hyperparameters. We compared the two methods under different computational cost.

Remarkably, we found in many cases phenomenon predicted by our theory matches with observations in practice. Our experiments suggested there is a trade-off between the computational cost and the convergence speed for underparameterized neural networks. SVRG outperformed SGD after a few epochs in this regime. Interestingly, in the case of overparameterized model, a setting that matches modern day neural networks training, SGD strictly dominated SVRG by showing a faster convergence throughout the entire training.

## 4.1 SIMULATIONS ON NOISY LEAST SQUARES REGRESSION MODEL

We first performed numerical simulations of the dynamics derived in Theorem 2 for SGD and Theorem 4 for SVRG. We picked a data distribution, with data dimension $d = 100$, and the spectrum of $\Sigma$ is given by an exponential decay schedule from 1 to 0.01. For both methods, we picked 50 learning rate from 1.5 to 0.01 using a exponential decay schedule. For SVRG, we further picked a set of snapshot intervals for each learning rate: $\{256, 128, 64\}$. We performed simulations in both underparameterized and overparameterized setting (namely with and without label noise), and plotted the lower bound curves over all hyperparameters at Figure 2. The $x$-axis represents the normalized total computational cost, denoting $tbN^{-1}$, which is equivalent to the notion of an epoch in finite dataset setting. And the loss in Figure 2 does not contain bayes error (i.e. $\frac{1}{2}\sigma_y^2$).

We have the following observations from our simulations. In the case with label noise, the plot demonstrated an explicit trade-off between computational cost and convergence speed. We observed a crossing point of between SGD and SVRG appear, indicating SGD achieved a faster convergence speed in the first phase of the training, but converged to a higher loss, for all per-iteration compute cost. Hence it shows that one can trade more compute cost for convergence speed by choosing SGD than SVRG, and vice versa. Interestingly, we found that the the per-iteration computational cost does not seem to affect the time crossing point takes place. For all these three costs, the crossing points in the plot are at around the same time: 5.5 epochs. In the case of no label noise, we observed both methods achieved linear convergence, while SGD achieved a much faster rate than SVRG, showing absolute dominance in this regime.

## 4.2 BENCHMARK DATASETS

In this section, we performed a similar investigation as in the last section, on two standard machine learning benchmark datasets: MNIST (LeCun et al., 1998) and CIFAR-10 (Krizhevsky, 2009). We present the results from underparameterized setting first, followed by the overparameterized setting. We performed experiments with three batchsizes for SGD: $\{64, 128, 256\}$, and an equivalent batchsize for SVRG. For each batch size, we pick 8 learning rates varying from 0.3 to 0.001 following an exponential schedule. Additionally, we chose three snapshot intervals for every computational budget, searching over the best snapshot interval given the data. Hence for each per-iteration computational cost $\{64, 128, 256\}$, there are 24 groups of experiments for SVRG and 8 groups of experiments for SGD.

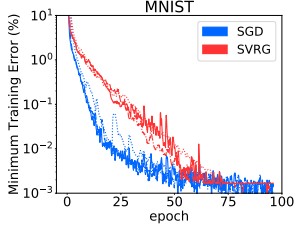 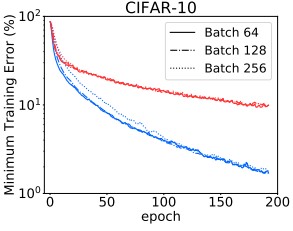

(a) Over-paremetrized MLP  (b) overparameterized CNN

Figure 4: The minimum loss achieved by following SGD (blue) and SVRG (red) over a set of hyperparameters for training on MNIST and CIFAR-10 with overparameterized models. In this setting we observed strict dominance of SGD over SVRG in convergence speed for all computational cost, matching our previous theoretical prediction.

### 4.2.1 UNDERPARAMETERIZED SETTING

For MNIST, we trained two underparameterized model: 1. logistic regression $784 - 10$ 2. a underparameterized two layer MLP $784 - 10 - 10$ where the hidden layer has 10 neurons. For CIFAR-10, we chose a underparameterized convolutional neural network model, which has only two 8-channel convolutional layers, and one 16-channel convolutional layer with one additional fully-connected layer. Filter size is 5. The lowest loss achieved over all hyperparameters for these models for each per-iteration computational cost are shown in Figure 3.

From these experiments, we observe that on MNIST, the results with underparameterized model were consistent with the dynamics simulation of noisy least squares regression model with label noise. First of all, SGD converged faster in the early phase, resulting in a crossing point between SGD and SVRG. It showed a trade-offs between computational cost and convergence speed: before the crossing point, SGD converged faster than SVRG; after crossing point, SVRG attained a lower loss. In addition, in Fig 3a, all the crossing points of three costs matched at the same epoch (around 5), which was also consistent with the our findings with noisy least squares regression model. On CIFAR-10, SGD achieved slightly faster convergence in the early phase, but was surpassed by SVRG around $17 - 25$ epochs, again showing a trade-off between compute and speed.

### 4.2.2 THE OVERPARAMETERIZED SETTING

Lastly, we compared SGD and SVRG on MNIST and CIFAR-10 using overparameterized models. For MNIST, we used a MLP with two hidden layers, each layer having 1024 neurons. For CIFAR-10, we chose a large convolutional network, which has one 64-channel convolutional layer, one 128-channel convolutional layer followed by one 3200 to 1000 fully connected layer and one 1000 to 10 fully connected layer.

The lowest loss achieved over all hyperparameters for these models for each per-iteration computational cost are shown in Figure 4. For training on MNIST, both SGD and SVRG attained close to zero training loss. The results were again consistent with our dynamics analysis on the noisy linear regression model without label noise. SGD has a strict advantage over SVRG, and achieved a much faster convergence rate than SVRG throughout the entire training. As for CIFAR-10, we stopped the training before either of the two got close to zero training loss due to lack of computing time. But we clearly see a trend of approaching to zero loss. Similarly, we also had the same observations as before, where SGD outperforms SVRG, confirms the limitation of variance reduction in the overparameterized regime.

## 5 DISCUSSION

In this paper, we studied the convergence properties of SGD and SVRG in the underparameterized and overparameterized settings. We provided a non-asymptotic analysis of both algorithms. We then investigated the question about which algorithm to use under certain total computational cost. We performed numerical simulations of dynamics equations for both methods, as well as extensive experiments on the standard machine learning datasets, MNIST and CIFAR-10. Remarkably, we found in many cases phenomenon predicted by our theory matched with observations in practice. Our experiments suggested there is a trade-off between the computational cost and the convergence speed for underparameterized neural networks. SVRG outperformed SGD after the first few epochs in this regime. In the case of overparameterized model, a setting that matches with modern day neural networks training, SGD strictly dominated SVRG by showing a faster convergence for all computational cost.

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

## A   APPENDIX

## B   LEMMA ABOUT GRADIENT COVARIANCE

**Lemma 6** (Gradient Covariance). *Given the noisy linear regression objective function (Eq. 3), under the assumption that $x \sim \mathcal{N}(\mathbf{0}, \Sigma)$ with $\Sigma$ diagonal and $\boldsymbol{\theta}^* = 0$, we have*

$$diag(\mathbb{E}[\boldsymbol{xx}^\top \boldsymbol{\theta}^{(t)}\boldsymbol{\theta}^{(t)^\top}\boldsymbol{xx}^\top]) = \Big( 2\Sigma^2 + diag(\Sigma)diag(\Sigma)^\top \Big)\mathbb{E}[\boldsymbol{\theta}^{(t)^{\circ 2}}]$$

$$\mathbb{E}[X_b X_b^\top \boldsymbol{\theta}^{(t)}\boldsymbol{\theta}^{(t)^\top} X_b X_b^\top] - \Sigma\mathbb{E}[\boldsymbol{\theta}^{(t)}\boldsymbol{\theta}^{(t)^\top}]\Sigma = \frac{1}{b}\Big( \mathbb{E}[\boldsymbol{xx}^\top\boldsymbol{\theta}^{(t)}\boldsymbol{\theta}^{(t)^\top}\boldsymbol{xx}^\top] - \Sigma\mathbb{E}[\boldsymbol{\theta}^{(t)}\boldsymbol{\theta}^{(t)^\top}]\Sigma\Big)$$

*Proof.* In the following proof, we define the entry-wise $p$ power on vector $\boldsymbol{x}$ as $\boldsymbol{x}^{\circ p}$. Under our assumption $\boldsymbol{\mu} = 0$, $\boldsymbol{\theta}^* = 0$ and $\Sigma$ diagonal, for $\boldsymbol{x} \sim \mathcal{N}(\mathbf{0}, \Sigma)$, $\boldsymbol{x} \in \mathbb{R}^d$, we have

$$\mathbb{E}_{\boldsymbol{x},\boldsymbol{\theta}^{(t)}}[\boldsymbol{xx}^\top\boldsymbol{\theta}^{(t)}\boldsymbol{\theta}^{(t)^\top}\boldsymbol{xx}^\top] = 2\Sigma^2\mathbb{E}[\boldsymbol{\theta}^{(t)}\boldsymbol{\theta}^{(t)^\top}] + \mathrm{Tr}(\Sigma\mathbb{E}[\boldsymbol{\theta}^{(t)}\boldsymbol{\theta}^{(t)^\top}])\Sigma. \tag{12}$$

Eq. 12 is a conclusion from The Matrix Cookbook (See section 8.2.3 in Petersen & Pedersen (2012)).

Then, for its main diagonal term, we have:

$$\mathrm{diag}(\mathbb{E}_{\boldsymbol{x},\boldsymbol{\theta}^{(t)}}[\boldsymbol{xx}^\top\boldsymbol{\theta}^{(t)}\boldsymbol{\theta}^{(t)^\top}\boldsymbol{xx}^\top]) = 2\Sigma^2\mathbb{E}[\boldsymbol{\theta}^{(t)^{\circ 2}}] + \mathrm{diag}(\Sigma)\mathrm{diag}(\Sigma)^\top\mathbb{E}[\boldsymbol{\theta}^{(t)^{\circ 2}}] \tag{13}$$

$$= \Big( 2\Sigma^2 + \mathrm{diag}(\Sigma)\mathrm{diag}(\Sigma)^\top \Big)\mathbf{m}(\boldsymbol{\theta}^{(t)}) \tag{14}$$

Hence, for $\mathrm{C}\Big(\mathrm{M}(\boldsymbol{\theta}^{(t)})\Big)$, we have:

$$\mathrm{diag}\Big(\mathrm{C}\Big(\mathrm{M}(\boldsymbol{\theta}^{(t)})\Big)\Big) = \Big( \Sigma^2 + \mathrm{diag}(\Sigma)\mathrm{diag}(\Sigma)^\top \Big)\mathbf{m}(\boldsymbol{\theta}^{(t)}) \tag{15}$$

which is the first conclusion of Theorem 2.

Notice, this conclusion can be generalized to any square matrix $A$ not only for $\mathbb{E}[\boldsymbol{\theta}^{(t)}\boldsymbol{\theta}^{(t)^\top}]$, i.e. for any square matrix $A \in \mathbb{R}^{d \times d}$, with $x \sim \mathcal{N}(\mathbf{0}, \Sigma)$ and $\Sigma$ diagonal, since

$$\mathbb{E}_{\boldsymbol{x}}[\boldsymbol{xx}^\top A\boldsymbol{xx}^\top] = 2\Sigma^2 A + \mathrm{Tr}(\Sigma A)\Sigma. \tag{16}$$

we have

$$\mathrm{diag}(\mathbb{E}_{\boldsymbol{x}}[\boldsymbol{xx}^\top A\boldsymbol{xx}^\top]) = 2\Sigma^2\mathrm{diag}(A) + \mathrm{diag}(\Sigma)\mathrm{diag}(\Sigma)^\top\mathrm{diag}(A) \tag{17}$$

$$= \Big( 2\Sigma^2 + \mathrm{diag}(\Sigma)\mathrm{diag}(\Sigma)^\top \Big)\mathrm{diag}(A) \tag{18}$$

For batch gradient $X_b X_b^\top \boldsymbol{\theta}^{(t)}$, we have

$$\mathbb{E}[X_b X_b^\top \boldsymbol{\theta}^{(t)}\boldsymbol{\theta}^{(t)^\top} X_b X_b^\top] = \frac{1}{b^2}\mathbb{E}\Big[ (\sum_{i \in [N]_b} \boldsymbol{x}_i\boldsymbol{x}_i^\top)\boldsymbol{\theta}^{(t)}\boldsymbol{\theta}^{(t)^\top}(\sum_{i \in [N]_b} \boldsymbol{x}_i\boldsymbol{x}_i^\top) \Big] \tag{19}$$

$$= \frac{1}{b^2}b\mathbb{E}[\boldsymbol{xx}^\top\boldsymbol{\theta}^{(t)}\boldsymbol{\theta}^{(t)}\boldsymbol{xx}^\top] + \frac{1}{b^2}(b^2 - b)\mathbb{E}[\boldsymbol{xx}^\top]\mathbb{E}[\boldsymbol{\theta}^{(t)}\boldsymbol{\theta}^{(t)}]\mathbb{E}[\boldsymbol{xx}^\top] \tag{20}$$

$$= \frac{1}{b}\mathbb{E}[\boldsymbol{xx}^\top\boldsymbol{\theta}^{(t)}\boldsymbol{\theta}^{(t)^\top}\boldsymbol{xx}^\top] + \frac{b-1}{b}\Sigma\mathbb{E}[\boldsymbol{\theta}^{(t)}\boldsymbol{\theta}^{(t)^\top}]\Sigma \tag{21}$$

where $[N]_b$ is the index set of $X_b$.

$\square$

## C   THE PROOF OF THEOREM 2

**Theorem 2.** *Given the noisy linear regression objective function (Eq. 3), under the assumption that* $x \sim \mathcal{N}(\mathbf{0}, \Sigma)$ *with* $\Sigma$ *diagonal and* $\boldsymbol{\theta}^* = 0$*, we can express* $\mathrm{C}(\mathrm{M}(\boldsymbol{\theta}))$ *as a function of* $\mathbf{m}(\boldsymbol{\theta})$*:*

$$diag\Big(\mathrm{C}\big(\mathrm{M}(\boldsymbol{\theta})\big)\Big) = \Big(\Sigma^2 + diag(\Sigma)diag(\Sigma)^\top\Big)\mathbf{m}(\boldsymbol{\theta})$$

*Then we derive following dynamics of expected second moment of* $\boldsymbol{\theta}$*:*

$$\mathbf{m}(\boldsymbol{\theta}^{(t)}) = R^t\Big(\mathbf{m}(\boldsymbol{\theta}^{(0)}) - \frac{(\mathrm{I} - R)^{-1}\boldsymbol{n}}{b}\Big) + \frac{(\mathrm{I} - R)^{-1}\boldsymbol{n}}{b},$$

*Proof.*

$$\boldsymbol{\theta}^{(t+1)}\boldsymbol{\theta}^{(t+1)\top} = (\mathbf{I} - \alpha X_b X_b^\top)\boldsymbol{\theta}^{(t)}\boldsymbol{\theta}^{(t)\top}(\mathbf{I} - \alpha X_b X_b^\top) + \frac{\alpha}{\sqrt{b}}(\mathbf{I} - \alpha X_b X_b^\top)\boldsymbol{\theta}^{(t)}\boldsymbol{\epsilon}_b^\top X_b^\top \quad (22)$$

$$+ \frac{\alpha}{\sqrt{b}}X_b\boldsymbol{\epsilon}_b\boldsymbol{\theta}^{(t)\top}(\mathbf{I} - \alpha X_b X_b^\top) + \frac{\alpha^2}{b}X_b\boldsymbol{\epsilon}_b\boldsymbol{\epsilon}_b^\top X_b^\top \quad (23)$$

Since, $\mathbb{E}[\boldsymbol{\epsilon}_b] = \mathbf{0}$, and $\boldsymbol{\epsilon}_b$ is independent with $\boldsymbol{\theta}^{(t)}, X_b$, we have:

$$\mathbb{E}[X_b\boldsymbol{\epsilon}_b\boldsymbol{\theta}^{(t)\top}(\mathbf{I} - \alpha X_b X_b^\top)] = \mathbf{0} \quad (24)$$

$$\mathbb{E}[(\mathbf{I} - \alpha X_b X_b^\top)\boldsymbol{\theta}^{(t)}\boldsymbol{\epsilon}_b^\top X_b^\top] = \mathbf{0} \quad (25)$$

and,

$$\mathbb{E}_{\boldsymbol{\epsilon}_b, X_b}\Big[\frac{\alpha^2}{b}X_b\boldsymbol{\epsilon}_b\boldsymbol{\epsilon}_b^\top X_b^\top\Big] = \frac{\alpha^2}{b}\mathbb{E}_{X_b}\Big[X_b\big(\mathbb{E}[\boldsymbol{\epsilon}_b\boldsymbol{\epsilon}_b^\top]\big)X_b\Big] \quad (26)$$

$$= \frac{\alpha^2}{b}\mathbb{E}_{X_b}\Big[X_b\big(\sigma_y^2\mathbf{I}\big)X_b\Big] \quad (27)$$

$$= \frac{\alpha^2\sigma_y^2}{b}\Sigma. \quad (28)$$

Since $X_b$ is independent with $\boldsymbol{\theta}^{(t)}$, we have:

$$\mathbb{E}\Big[(\mathbf{I} - \alpha X_b X_b^\top)\boldsymbol{\theta}^{(t)}\boldsymbol{\theta}^{(t)\top}(\mathbf{I} - \alpha X_b X_b^\top)\Big] = (\mathbf{I} - \alpha\Sigma)\mathbb{E}[\boldsymbol{\theta}^{(t)}\boldsymbol{\theta}^{(t)\top}](\mathbf{I} - \alpha\Sigma) \quad (29)$$

$$+ \alpha^2\Big(\mathbb{E}[X_b X_b^\top\boldsymbol{\theta}^{(t)}\boldsymbol{\theta}^{(t)\top}X_b X_b^\top] - \Sigma\mathbb{E}[\boldsymbol{\theta}^{(t)}\boldsymbol{\theta}^{(t)\top}]\Sigma\Big). \quad (30)$$

Thus,

$$\mathbb{E}[\boldsymbol{\theta}^{(t+1)}\boldsymbol{\theta}^{(t+1)\top}] = (\mathbf{I} - \alpha\Sigma)\mathbb{E}[\boldsymbol{\theta}^{(t)}\boldsymbol{\theta}^{(t)\top}](\mathbf{I} - \alpha\Sigma) + \frac{\alpha^2}{b}(\mathbb{E}[\boldsymbol{x}\boldsymbol{x}^\top\boldsymbol{\theta}^{(t)}\boldsymbol{\theta}^{(t)\top}\boldsymbol{x}\boldsymbol{x}^\top] \quad (31)$$

$$- \Sigma\mathbb{E}[\boldsymbol{\theta}^{(t)}\boldsymbol{\theta}^{(t)\top}]\Sigma) + \frac{\alpha^2\sigma_y^2}{b}\Sigma \quad (32)$$

$$= (\mathbf{I} - \alpha\Sigma)\mathbb{E}[\boldsymbol{\theta}^{(t)}\boldsymbol{\theta}^{(t)\top}](\mathbf{I} - \alpha\Sigma) + \frac{\alpha^2}{b}\mathrm{C}(\mathrm{M}(\boldsymbol{\theta}^{(t)})) + \frac{\alpha^2\sigma_y^2}{b}\Sigma \quad (33)$$

For its diagonal term, we have:

$$\mathbf{m}(\boldsymbol{\theta}^{(t+1)}) = diag(\mathbb{E}[\boldsymbol{\theta}^{(t+1)}\boldsymbol{\theta}^{(t+1)\top}]) \quad (34)$$

$$= \Big((\mathbf{I} - \alpha\Sigma)^2 + \frac{\alpha^2}{b}(\Sigma^2 + diag(\Sigma)diag(\Sigma)^\top)\Big)\mathbf{m}(\boldsymbol{\theta}^{(t)}) + \frac{\alpha^2\sigma_y^2}{b}diag(\Sigma) \quad (35)$$

$$= R \cdot \mathbf{m}(\boldsymbol{\theta}^{(t)}) + \frac{1}{b}\boldsymbol{n} \quad (36)$$

This formula can be written as :

$$\mathbf{m}(\boldsymbol{\theta}^{(t+1)}) + b^{-1}(R - \mathrm{I})^{-1}\boldsymbol{n} = R\Big(\mathbf{m}(\boldsymbol{\theta}^{(t)}) + b^{-1}(R - \mathrm{I})^{-1}\boldsymbol{n}\Big) \tag{37}$$

$$\mathbf{m}(\boldsymbol{\theta}^{(t+1)}) = R^{t+1}\Big(\mathbf{m}(\theta_0) + b^{-1}(R - \mathrm{I})^{-1}\boldsymbol{n}\Big) - b^{-1}(R - \mathrm{I})^{-1}\boldsymbol{n}, \tag{38}$$

where

$$R = (\mathrm{I} - \alpha\Sigma)^2 + \alpha^2 b^{-1}(\Sigma^2 + \mathrm{diag}(\Sigma)\mathrm{diag}(\Sigma)^\top), \quad \boldsymbol{n} = \alpha^2\sigma_y^2\mathrm{diag}(\Sigma). \tag{39}$$

$\square$

## D  THE PROOF OF LEMMA 3

**Lemma 3.** *The dynamics of the second moment of the iterate following SVRG update rule is given by,*

$$\mathrm{M}(\boldsymbol{\theta}^{(mT+t+1)}) = \underbrace{(\mathrm{I} - \alpha\Sigma)\mathrm{M}(\boldsymbol{\theta}^{(mT+t)})(\mathrm{I} - \alpha\Sigma)}_{\text{① gradient descent shrinkage}} + \underbrace{\frac{\alpha^2}{b}\mathrm{C}\Big(\mathrm{M}(\boldsymbol{\theta}^{(mT+t)})\Big)}_{\text{② input noise}} + \underbrace{\frac{\alpha^2\sigma_y^2}{N}\Sigma}_{\text{③ label noise}}$$

$$+ \underbrace{\alpha^2\frac{N+b}{Nb}\mathrm{C}\Big(\mathrm{M}(\boldsymbol{\theta}^{(mT)})\Big)}_{\text{④variance due to } \tilde{\mathbf{g}}^{(mT+t)}} \underbrace{-\frac{\alpha^2}{b}\Big(\mathrm{C}\Big(\mathrm{M}(\boldsymbol{\theta}^{(mT)})P^t\Big) + \mathrm{C}\Big(P^t\mathrm{M}(\boldsymbol{\theta}^{(mT)})\Big)\Big)}_{\text{⑤ Variance reduction from control variate}}$$

*Proof.* For SVRG update rule Eq. 8, we have:

$$\boldsymbol{\theta}^{(mT+t+1)} = \left(\mathrm{I} - \alpha X_b X_b^\top\right)\boldsymbol{\theta}^{(mT+t)} + \alpha\left(X_b X_b^\top - X_N X_N^\top\right)\boldsymbol{\theta}^{(mT)} + \frac{\alpha}{\sqrt{N}}X_N\boldsymbol{\epsilon}_N \tag{40}$$

Using the update rule of SVRG, we can get the outer product of parameters as:

$$\boldsymbol{\theta}^{(mT+t+1)}\boldsymbol{\theta}^{(mT+t+1)\top} \tag{41}$$

$$= (\mathrm{I} - \alpha X_b X_b^\top)\boldsymbol{\theta}^{(mT+t)}\boldsymbol{\theta}^{(mT+t)\top}(\mathrm{I} - \alpha X_b X_b^\top) \tag{42}$$

$$+ \alpha(\mathrm{I} - \alpha X_b X_b^\top)\boldsymbol{\theta}^{(mT+t)}\boldsymbol{\theta}^{(mT)\top}(X_b X_b^\top - X_N X_N^\top) \tag{43}$$

$$+ \alpha(X_b X_b^\top - X_N X_N^\top)\boldsymbol{\theta}^{(mT)}\boldsymbol{\theta}^{(mT+t)\top}(\mathrm{I} - \alpha X_b X_b^\top) \tag{44}$$

$$+ \alpha^2(X_b X_b^\top - X_N X_N^\top)\boldsymbol{\theta}^{(mT)}\boldsymbol{\theta}^{(mT)\top}(X_b X_b^\top - X_N X_N^\top) \tag{45}$$

$$+ \frac{\alpha}{\sqrt{N}}X_N\boldsymbol{\epsilon}_N\boldsymbol{\theta}^{(mT+t)\top}(\mathrm{I} - \alpha X_b X_b^\top) + \frac{\alpha^2}{\sqrt{N}}X_N\boldsymbol{\epsilon}_N\boldsymbol{\theta}^{(mT)\top}(X_b X_b^\top - X_N X_N^\top) \tag{46}$$

$$+ \frac{\alpha}{\sqrt{N}}(\mathrm{I} - \alpha X_b X_b^\top)\boldsymbol{\theta}^{(mT+t)}\boldsymbol{\epsilon}_N^\top X_N^\top + \frac{\alpha^2}{\sqrt{N}}(X_b X_b^\top - X_N X_N^\top)\boldsymbol{\theta}^{(mT)}\boldsymbol{\epsilon}_N^\top X_N^\top \tag{47}$$

$$+ \frac{\alpha^2}{N}X_N\boldsymbol{\epsilon}_N\boldsymbol{\epsilon}_N^\top X_N^\top \tag{48}$$

Likewise, since $\mathbb{E}[\boldsymbol{\epsilon}_N] = \mathbf{0}$ and $\boldsymbol{\epsilon}_N$ is independent with $X_b$, $X_N$ and $\boldsymbol{\theta}^{(t)}$, we have the expectation of equation 46, equation 47 equal to $\mathbf{0}$. And same as SGD, we also have

$$\mathbb{E}_{\boldsymbol{\epsilon}_N, X_N}[X_N\boldsymbol{\epsilon}_N\boldsymbol{\epsilon}_N^\top X_N^\top] = \mathbb{E}_{X_N}\left[X_N\mathbb{E}_{\boldsymbol{\epsilon}_N}[\boldsymbol{\epsilon}_N\boldsymbol{\epsilon}_N^\top]X_N^\top\right] \tag{49}$$

$$= \mathbb{E}\left[X_N\Big(\sigma_y^2\mathrm{I}\Big)X_N^\top\right] \tag{50}$$

$$= \sigma_y^2\Sigma \tag{51}$$

Then, we give a significant formula about the expectation of $\boldsymbol{\theta}^{(mT+t)}\boldsymbol{\theta}^{(mT)}$, utilized to derive the expected term related to variance reduction amount.

$$\boldsymbol{\theta}^{(mT+t+1)}\boldsymbol{\theta}^{(mT)\top} = (\mathrm{I} - \alpha X_b X_b^\top)\boldsymbol{\theta}^{(mT+t)}\boldsymbol{\theta}^{(mT)\top} \tag{52}$$

$$+ \alpha(X_b X_b^\top - X_N X_N^\top)\boldsymbol{\theta}^{(mT)}\boldsymbol{\theta}^{(mT)\top} + \frac{\alpha}{\sqrt{N}}X_N\boldsymbol{\epsilon}_N\boldsymbol{\theta}^{(mT)\top} \tag{53}$$

Since $\mathbb{E}[X_N X_N^\top] = \mathbb{E}[X_b X_b^\top] = \Sigma$, and $\epsilon_N$ is independent with $X_N$ and $\boldsymbol{\theta}^{(mT)}$, the expectation of Eq. 53 is equal to $\mathbf{0}$. Therefore,

$$\mathbb{E}[\boldsymbol{\theta}^{(mT+t+1)}\boldsymbol{\theta}^{(mT)\top}] = (\mathbf{I} - \alpha\Sigma)\mathbb{E}[\boldsymbol{\theta}^{(mT+t)}\boldsymbol{\theta}^{(mT)\top}] = (\mathbf{I} - \alpha\Sigma)^{t+1}\mathbb{E}[\boldsymbol{\theta}^{(mT)}\boldsymbol{\theta}^{(mT)\top}] \quad (54)$$

$$= P^{t+1}\mathrm{M}(\boldsymbol{\theta}^{(mT)}) \quad (55)$$

which suggests the covariance between $\hat{\mathbf{g}}^{(mT+t)}$ and $\tilde{\mathbf{g}}^{(mT+t)}$ is exponentially decayed.

For every other term appearing in Eq. 41, we have the following conclusions. First, similar with SGD, we have the formula about gradient descent shrinkage as:

$$\mathbb{E}_{\boldsymbol{x},\boldsymbol{\theta}}[(\mathrm{I} - \alpha X_b X_b^\top)\boldsymbol{\theta}^{(mT+t)}\boldsymbol{\theta}^{(mT+t)\top}(\mathrm{I} - \alpha X_b X_b^\top)] \quad (56)$$

$$= (\mathrm{I} - \alpha\Sigma)\mathbb{E}[\boldsymbol{\theta}^{(mT+t)}\boldsymbol{\theta}^{(mT+t)\top}](\mathrm{I} - \alpha\Sigma) \quad (57)$$

$$+ \alpha^2\left(\mathbb{E}[X_b X_b^\top \boldsymbol{\theta}^{(mT+t)}\boldsymbol{\theta}^{(mT+t)\top} X_b X_b^\top] - \Sigma\mathbb{E}[\boldsymbol{\theta}^{(mT+t)}\boldsymbol{\theta}^{(mT+t)\top}]\Sigma\right) \quad (58)$$

$$= (\mathrm{I} - \alpha\Sigma)\mathbb{E}[\boldsymbol{\theta}^{(mT+t)}\boldsymbol{\theta}^{(mT+t)\top}](\mathrm{I} - \alpha\Sigma) \quad (59)$$

$$+ \alpha^2 b^{-1}\left(\mathbb{E}_{\boldsymbol{x}}[\boldsymbol{x}\boldsymbol{x}^\top \mathrm{M}(\boldsymbol{\theta}^{(mT+t)})\boldsymbol{x}\boldsymbol{x}^\top] - \Sigma\mathrm{M}(\boldsymbol{\theta}^{(mT+t)})\Sigma\right) \quad (60)$$

Using Eq. 54, we have following conclusion for variance reduction term from control variate. We first take expectation over $\boldsymbol{\theta}^{(mT+t)}\boldsymbol{\theta}^{(mT)\top}$ with Eq. 54 due to the independence among $X_b, X_N$ and $\boldsymbol{\theta}$.

$$\mathbb{E}\left[\alpha(\mathrm{I} - \alpha X_b X_b^\top)\boldsymbol{\theta}^{(mT+t)}\boldsymbol{\theta}^{(mT)\top}(X_b X_b^\top - X_N X_N^\top)\right] \quad (61)$$

$$= \mathbb{E}_{X_b, X_N}\left[\alpha(\mathrm{I} - \alpha X_b X_b^\top)P^t\mathrm{M}(\boldsymbol{\theta}^{(mT)})(X_b X_b^\top - X_N X_N^\top)\right] \quad (62)$$

$$= -\alpha^2 b^{-1}\left(\mathbb{E}_{\boldsymbol{x}}[\boldsymbol{x}\boldsymbol{x}^\top P^t\mathrm{M}(\boldsymbol{\theta}^{(mT)})\boldsymbol{x}\boldsymbol{x}^\top] - \Sigma P^t\mathrm{M}(\boldsymbol{\theta}^{(mT)})\Sigma\right) \quad (63)$$

$$\mathbb{E}\left[\alpha(X_b X_b^\top - X_N X_N^\top)\boldsymbol{\theta}^{(mT)}\boldsymbol{\theta}^{(mT+t)\top}(\mathrm{I} - \alpha X_b X_b^\top)\right] \quad (64)$$

$$= \mathbb{E}_{X_b, X_N}\left[\alpha(X_b X_b^\top - X_N X_N^\top)\mathrm{M}(\boldsymbol{\theta}^{(mT)})P^t(\mathrm{I} - \alpha X_b X_b^\top)\right] \quad (65)$$

$$= -\alpha^2 b^{-1}\left(\mathbb{E}_{\boldsymbol{x}}[\boldsymbol{x}\boldsymbol{x}^\top \mathrm{M}(\boldsymbol{\theta}^{(mT)})P^t\boldsymbol{x}\boldsymbol{x}^\top] - \Sigma\mathrm{M}(\boldsymbol{\theta}^{(mT)})P^t\Sigma\right) \quad (66)$$

For the forth term, which represents the variance of $\tilde{\mathbf{g}}^{(mT+t)}$, we consider the independence between $X_b$ and $X_N$ and get

$$\mathbb{E}\left[\alpha^2(X_b X_b^\top - X_N X_N^\top)\boldsymbol{\theta}^{(mT)}\boldsymbol{\theta}^{(mT)\top}(X_b X_b^\top - X_N X_N^\top)\right] \quad (67)$$

$$= \alpha^2\frac{N+b}{Nb}\left(\mathbb{E}_{\boldsymbol{x}}[\boldsymbol{x}\boldsymbol{x}^\top \mathrm{M}(\boldsymbol{\theta}^{(mT)})\boldsymbol{x}\boldsymbol{x}^\top] - \Sigma\mathrm{M}(\boldsymbol{\theta}^{(mT)})\Sigma\right) \quad (68)$$

Thus,

$$\mathrm{M}(\boldsymbol{\theta}^{(mT+t+1)}) = \left(\mathrm{I} - \alpha\Sigma\right)\mathrm{M}(\boldsymbol{\theta}^{(mT+t)})\left(\mathrm{I} - \alpha\Sigma\right) \quad (69)$$

$$+ \alpha^2 b^{-1}\left(\mathbb{E}_{\boldsymbol{x}}[\boldsymbol{x}\boldsymbol{x}^\top \mathrm{M}(\boldsymbol{\theta}^{(mT+t)})\boldsymbol{x}\boldsymbol{x}^\top] - \Sigma\mathrm{M}(\boldsymbol{\theta}^{(mT+t)})\Sigma\right) \quad (70)$$

$$+ \alpha^2\frac{N+b}{Nb}\left(\mathbb{E}_{\boldsymbol{x}}[\boldsymbol{x}\boldsymbol{x}^\top \mathrm{M}(\boldsymbol{\theta}^{(mT)})\boldsymbol{x}\boldsymbol{x}^\top] - \Sigma\mathrm{M}(\boldsymbol{\theta}^{(mT)})\Sigma\right) \quad (71)$$

$$- \alpha^2 b^{-1}\left(\mathbb{E}_{\boldsymbol{x}}[\boldsymbol{x}\boldsymbol{x}^\top \mathrm{M}(\boldsymbol{\theta}^{(mT)})P^t\boldsymbol{x}\boldsymbol{x}^\top] - \Sigma\mathrm{M}(\boldsymbol{\theta}^{(mT)})P^t\Sigma\right) \quad (72)$$

$$- \alpha^2 b^{-1}\left(\mathbb{E}_{\boldsymbol{x}}[\boldsymbol{x}\boldsymbol{x}^\top P^t\mathrm{M}(\boldsymbol{\theta}^{(mT)})\boldsymbol{x}\boldsymbol{x}^\top] - \Sigma P^t\mathrm{M}(\boldsymbol{\theta}^{(mT)})\Sigma\right) \quad (73)$$

$$+ \frac{\alpha^2\sigma_y^2}{N}\Sigma \quad (74)$$

Under our definition, it can be expressed as:

$$M(\boldsymbol{\theta}^{(mT+t+1)}) = \underbrace{(I - \alpha\Sigma)M(\boldsymbol{\theta}^{(mT+t)})(I - \alpha\Sigma)}_{\text{① gradient descent shrinkage}} + \underbrace{\frac{\alpha^2}{b}C\Big(M(\boldsymbol{\theta}^{(mT+t)})\Big)}_{\text{② input noise}} + \underbrace{\frac{\alpha^2\sigma_y^2}{N}\Sigma}_{\text{③ label noise}} \tag{75}$$

$$+ \underbrace{\alpha^2\frac{N+b}{Nb}C\Big(M(\boldsymbol{\theta}^{(mT)})\Big)}_{\text{④ variance due to } \tilde{\mathbf{g}}^{(mT+t)}} \underbrace{-\frac{\alpha^2}{b}\Big(C\Big(M(\boldsymbol{\theta}^{(mT)})P^t\Big) + C\Big(P^tM(\boldsymbol{\theta}^{(mT)})\Big)\Big)}_{\text{⑤ Variance reduction from control variate}}$$

$\square$

# E  THE PROOF OF THEOREM 4

**Theorem 4.** *Given the noisy linear regression objective function (Eq. 3), under the assumption that $x \sim \mathcal{N}(\mathbf{0}, \Sigma)$ with $\Sigma$ diagonal and $\boldsymbol{\theta}^* = 0$, the dynamics for SVRG in $\mathbf{m}(\boldsymbol{\theta})$ is given by:*

$$\mathbf{m}(\boldsymbol{\theta}^{((m+1)T)}) = \lambda(\alpha, b, T, N, \Sigma)\mathbf{m}(\boldsymbol{\theta}^{(mT)}) + (I - R^T)(I - R)^{-1}\frac{\mathbf{n}}{N},$$

$$\lambda(\alpha, b, T, N, \Sigma) = R^T - \Big(\sum_{k=0}^{T-1} R^k Q P^{-k}\Big)P^{T-1} + (I - R^T)(I - R)^{-1}F$$

*Proof.* Form lemma 3 and lemma 6, we can get:

$$\mathbf{m}(\boldsymbol{\theta}^{(mT+t+1)}) = R\mathbf{m}(\boldsymbol{\theta}^{(mT+t)}) - QP^t\mathbf{m}(\boldsymbol{\theta}^{(mT)}) + F\mathbf{m}(\boldsymbol{\theta}^{(mT)}) + N^{-1}\mathbf{n} \tag{76}$$

where

$$R = (I - \alpha\Sigma)^2 + \frac{\alpha^2}{b}(\Sigma^2 + \text{diag}(\Sigma)\text{diag}(\Sigma)^\top) \quad Q = \frac{2\alpha^2}{b}(\Sigma^2 + \text{diag}(\Sigma)\text{diag}(\Sigma)^\top), \tag{77}$$

$$F = \frac{\alpha^2(N+b)}{Nb}(\Sigma^2 + \text{diag}(\Sigma)\text{diag}(\Sigma)^\top), \quad P = I - \alpha\Sigma, \quad \mathbf{n} = \alpha^2\sigma_y^2 diag(\Sigma). \tag{78}$$

Recursively expending the above formula from $\mathbf{m}(\boldsymbol{\theta}^{((m+1)T)})$ to $\mathbf{m}(\boldsymbol{\theta}^{(mT)})$, we can get the following result:

$$\mathbf{m}(\boldsymbol{\theta}^{((m+1)T)}) \tag{79}$$

$$= R\Big(R\mathbf{m}(\boldsymbol{\theta}^{(mT+T-2)}) - QP^{T-2}\mathbf{m}(\boldsymbol{\theta}^{(mT)}) + F\mathbf{m}(\boldsymbol{\theta}^{(mT)}) + N^{-1}\mathbf{n}\Big) \tag{80}$$

$$- QP^{T-1}\mathbf{m}(\boldsymbol{\theta}^{(mT)}) + F\mathbf{m}(\boldsymbol{\theta}^{(mT)}) + N^{-1}\mathbf{n} \tag{81}$$

$$= R^2\mathbf{m}(\boldsymbol{\theta}^{(mT+T-2)}) - \Big(\sum_{k=0}^{1} R^k Q P^{-k}\Big)P^{T-1}\mathbf{m}(\boldsymbol{\theta}^{(mT)}) \tag{82}$$

$$+ \Big(\sum_{k=0}^{1} R^k\Big)\Big(F\mathbf{m}(\boldsymbol{\theta}^{(mT)}) + N^{-1}\mathbf{n}\Big) \tag{83}$$

$$= \cdots \tag{84}$$

$$= R^T\mathbf{m}(\boldsymbol{\theta}^{(mT)}) - \Big(\sum_{k=0}^{T-1} R^k Q P^{-k}\Big)P^{T-1}\mathbf{m}(\boldsymbol{\theta}^{(mT)}) \tag{85}$$

$$+ \Big(\sum_{k=0}^{T-1} R^k\Big)\Big(F\mathbf{m}(\boldsymbol{\theta}^{(mT)}) + N^{-1}\mathbf{n}\Big) \tag{86}$$

$$= R^T\mathbf{m}(\boldsymbol{\theta}^{(mT)}) - \Big(\sum_{k=0}^{T-1} R^k Q P^{-k}\Big)P^{T-1}Q\mathbf{m}(\boldsymbol{\theta}^{(mT)}) \tag{87}$$

$$+ (I - R^T)(I - R)^{-1}\Big(F\mathbf{m}(\boldsymbol{\theta}^{(mT)}) + N^{-1}\mathbf{n}\Big) \tag{88}$$

In other word, Eq. 79 describe the dynamic of expected second moment of iterate between two nearby snapshots,

$$\mathbf{m}(\boldsymbol{\theta}^{((m+1)T)}) = \lambda(\alpha, b, T, N, \Sigma)\mathbf{m}(\boldsymbol{\theta}^{(mT)}) + (\mathbf{I} - R^T)(\mathbf{I} - R)^{-1}\frac{\boldsymbol{n}}{N}, \tag{89}$$

where

$$\lambda(\alpha, b, T, N, \Sigma) = R^T - \Big(\sum_{k=0}^{T-1} R^k Q P^{-k}\Big) P^{T-1} + (\mathbf{I} - R^T)(\mathbf{I} - R)^{-1}F \tag{90}$$

Since $\mathrm{I} - R$ and $\mathrm{I} - R^T$ are commutable, $\frac{\mathrm{I}-R^T}{\mathrm{I}-R} = (\mathrm{I} - R)^{-1}(\mathrm{I} - R^T) = (\mathrm{I} - R^T)(\mathrm{I} - R)^{-1}$

$\square$

## F   THE PROOF OF COROLLARY 5

**Corollary 5.** *Without the label noise, the dynamics of the second moment following SGD is given by,*

$$\mathbf{m}(\boldsymbol{\theta}^{(t)}) = R^t \mathbf{m}(\boldsymbol{\theta}^{(0)}),$$

*and the dynamics of SVRG is given by,*

$$\mathbf{m}(\boldsymbol{\theta}^{((m+1)T)}) = \lambda(\alpha, b, T, N, \Sigma)\mathbf{m}(\boldsymbol{\theta}^{(mT)}),$$

*where $\lambda$ is defined in Eq.( 11).*

*Proof.* If without label noise, i.e. $\sigma_y^2 = 0$, we can directly draw the corollary for the setting without label noise, based on Theorem 2 and Theorem 4. Setting $\sigma_y^2 = 0$, we can draw:

$$\mathbf{m}(\boldsymbol{\theta}^{(t)}) = R^t \mathbf{m}(\boldsymbol{\theta}^{(0)}),$$

and the dynamics of SVRG is given by,

$$\mathbf{m}(\boldsymbol{\theta}^{((m+1)T)}) = \lambda(\alpha, b, T, \Sigma, N)\mathbf{m}(\boldsymbol{\theta}^{(mT)}),$$

where $\lambda$ is defined in Eq.( 11). $\square$

## G   THE SENSITIVITY OF $N$

In our theoretical analysis (Section 3), we evaluate a large batch gradient $\bar{\mathbf{g}}$ to control variance. That is because any number of data points can be directly sampled form the true distribution. But in order to compare the computational cost between SVRG and SGD, we set the number of data points used to calculate $\bar{\mathbf{g}}$ as $N$, which is slightly different with the original SVRG's setup of full-batch gradient. Therefore, we evaluate the sensitivity of $N$ to illustrate when $N$ is beyond a threshold, it will cause little difference in convergence speed for SVRG.

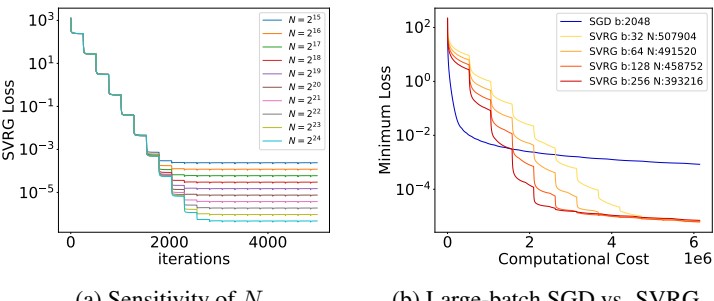

(a) Sensitivity of $N$           (b) Large-batch SGD vs. SVRG

Figure 5: Evaluate the sensitivity of $N$ in Figure 5a with $\alpha = 0.5$, $T = 256$ and $b = 64$ for SVRG under the noisy least square model. And compare large batch SGD to SVRG in Figure 5b with $T = 256$ and fixing computational budget as 2048, varying $b', N$ for SVRG.

From figure 5a, we can tell $N$ has little effect on the convergence speed of SVRG under the noisy least square model, but it determines the constant term of label noise in Eq. 9 which determines the level of final loss.

Besides, we also compare large batch SGD to SVRG in Figure 5b under the computation budget $b = 2048$ with fixed snapshot interval $T = 256$ for SVRG, expentionally picking 50 learning rates from 1.5 to 0.01, varying $b'$, $N$ according to $b' = \frac{1}{2}(1 - \frac{N}{Tb})b$.

