# OpenReview forum: "A Non-asymptotic comparison of SVRG and SGD: tradeoffs between compute and speed"
_ICLR.cc/2020/Conference — Reject_

### Official Review · AnonReviewer1 · 2019-10-07
**Official Blind Review #1**

**Rating:** 3

**Review:**

This paper examines the tradeoffs between applying SVRG and SGD for training neural networks by providing an analysis of noisy least squares regression problems as well as experiments on simple MLPs and CNNs on MNIST and CIFAR-10. The theory analyzes a linear model where both the input $x$ and label noise $\epsilon$ follow Gaussian distributions. Under these assumptions, the paper shows that SVRG is able to converge to a smaller neighborhood at a slower rate than SGD, which converges faster to a larger neighborhood. This analysis coincides with the experimental behavior applied to neural networks, where one observes when training underparameterized models that SGD significantly outperforms SVRG initially, but SVRG is able to attain a lower loss value asymptotically. In the overparameterized regime, SGD is demonstrated to always outperform SVRG experimentally, which is argued to coincide with the case where there is no label noise in the theory.

Strengths:

I liked how the authors distinguished between the underparameterized and overparameterized regimes in the analysis and experiments. This allowed them to observe different behavior between the two regimes when comparing SVRG and SGD. I also found the authors' setting of analyzing noisy least squares problems to be interesting because of its potential usefulness for both analytically and empirically understanding certain forms of DL phenomena. The introduction is also well-written.

Weaknesses:

One aspect that I found unclear about the paper is its definition of the SVRG algorithm. In the analysis, the paper examines the expected risk least squares problem, and (if I understand correctly), considers the version of SVRG where the snapshot gradient is sampled i.i.d. over a large batch randomly from the true distribution. This is in contrast to the original SVRG method, which was designed for the empirical risk (or finite-sum) problem, where the set of datapoints is fixed. This coincides with the experiments, where the full training set is used to evaluate the snapshot gradient. Is this the correct interpretation of the theoretical and experimental results?

If so, how does this theoretical version of SVRG compare to a stochastic gradient method with large batch size? Does the theoretical behavior and insights exhibited by SVRG differ significantly from the theoretical behavior of SGD with larger batch size?

In addition, is the noisy least squares regression model with a diagonal data covariance equivalent to a separable quadratic problem? If so, it may not be surprising that the expected second moment of each parameter would evolve independently from each other, as noted at the end of Section 2.

I also found some of the theorems and proofs difficult to follow. This is partly due to some inconsistent notation: what is $B$ (vs $b$) (pg. 4)? Is $A = M$ (pg. 4)? What does $\circ$ denote in the exponents in the Appendix? What is the meaning of the constants defined in Definition 1? Some further explanation of the theoretical results (such as the meaning of those constants and more directly comparing the bounds for SVRG and SGD) would help with interpreting their results, particularly Theorem 4.

Along these lines, is it true that the rate of convergence for SGD is faster than the rate for SVRG? The constants made this difficult to tell, and no explanation was provided (although this was claimed in the Experiments section).

Most steps in the proof were also left unexplained, which made it difficult to follow without knowledge of certain properties of multivariate Gaussians. Some necessary assumptions were also missing from the definition of the model; in particular, the paper did not specify the relationship between $\epsilon_i$ and $x_i$ (which I assume are independent).

The experiments could also certainly be reinforced with some larger scale experiments on some larger datasets (such as ImageNet). One could see that the results became much more messy in the case of the underparameterized CNN on CIFAR-10 for example, and I wonder if this phenomena still holds with much larger datasets.

Some additional typos:
- Should use \citep for the Johnson & Zhang reference at the end of page 3
- SVRG Dynamics and Decay "R"ate in page 5

Overall, although the paper provides an interesting observation and direction in contrasting the underparameterized and overparameterized regimes when comparing SVRG and SGD for training DNNs, in my opinion, the paper needs some additional refining, particularly in terms of clarity with respect to the theory and notation, and perhaps some more experiments. If I understand the theoretical and empirical SVRG algorithms correctly, I'm not currently convinced that the paper provides substantially more theoretical insight than before due to differences between the theoretical and empirical SVRG methods applied in this paper and the theoretical algorithm's similarity to large-batch SGD. The observation in the underparameterized regime, for example, has been highlighted in prior work even with logistic regression (particularly due to the cost of evaluating a full gradient), and a theoretical comparison of small-batch SGD and large-batch SGD neighborhood results for strongly convex problems (see Bottou, Curtis, and Nocedal (2018), for example) would lead to similar conclusions. Because of these reasons, I do not recommend this paper for publication at this time.

**Experience Assessment:**

I have published one or two papers in this area.

**Review Assessment: Checking Correctness Of Derivations And Theory:**

I assessed the sensibility of the derivations and theory.

**Review Assessment: Checking Correctness Of Experiments:**

I carefully checked the experiments.

**Review Assessment: Thoroughness In Paper Reading:**

I read the paper thoroughly.

---

> ### Author Response · Authors · 2019-11-15
> **Large batch SGD is not the same as SVRG. Please consider updating the score.**
>
> Thanks for your constructive feedback. We have revised our paper based on your suggestions.
>
> Q: The theoretical version of SVRG.
> >>  We compares SVRG and SGD under the same computational budget, and hence it is quite crucial in our theoretical analysis to use a snapshot gradient that requires a finite computational budget. We hence use a large batch of size N instead of the true gradient (which requires infinite data points). This does not change the validity of the results. We show this by adding a new experiment  in Appendix G, where we fix other hyperparameters (learning rate, snapshot interval), and only varied N. We observe for SGD, the loss at each timesteps before convergence is the same, and different N only resulted in a different convergence point. Since the loss at each timesteps was not affected by N, SGD still achieved lower loss in the first phase of the training than SVRG, and our conclusion is hence not affected by N.
>
> Q: large batch SGD?
> >> The work of large batch SGD is not relevant at all. The SVRG algorithm only requires a large batch gradient computation for a snapshot gradient, which is computed once every snapshot interval. In contrast, the large batch SGD computes large batch gradient at each time step. Therefore, these two algorithms demand a different analysis.
>
> Q: In addition, is the noisy least squares regression model with a diagonal data covariance equivalent to a separable quadratic problem?
> >> One mistake in our analysis has been found and fixed in our new version of paper. And all simulations were rerun based on new theorems, and our previous conclusions still hold. In  our new theorems, the coefficient matrix in the first equation of Theorem 2 changes from diagonal matrix to positive definite symmetric matrix. In this case, the second moment of each parameter no longer evolve independently.
>
> Q: What is $ B $ (vs $ b $ ) (pg. 4)? Is $A=M$ (pg. 4)? What does $ \circ $ denote in the exponents in the Appendix? What is the meaning of the constants defined in Definition 1? Some further explanation of the theoretical results (such as the meaning of those constants and more directly comparing the bounds for SVRG and SGD) would help with interpreting their results, particularly Theorem 4.
> >> We are sorry for the confusion caused by our unclear notations. In our new version of paper, we have cleaned up all inconsistent notations with replacing $B$ with $ b $, $ A $ with $ M $. $ \circ p $ appearing in appendix denotes element-wise p power on given matrix or vector, whose definition is now added in Appendix B.
> As for comparing the bounds, we were not trying to derive the bounds for SVRG and SGD but their exact expected loss (Theorem 4) after k steps given a specific group of hyperparameters. Then, under a fixed computational budget, we non-asymptotically compared SVRG with SGD in our numerical experiments (Section 5.1) based on Theorem 4. And the constants in Definition 1 are here for making our formulas more concise.
>
> Q: Along these lines, is it true that the rate of convergence for SGD is faster than the rate for SVRG?
> >> We numerically studied the exact expected loss derived for SVRG and SGD in Section 5.1 rather than comparing the convergence rates directly. Under the fixed computational budget, we compare SVRG and SGD over a wide-range and dense-sampled hyperparameters by drawing the minimum loss of them. The numerical experiments suggest SGD has a faster convergence speed. In the non-interpolation case (underparameterized), SGD is faster in the first phase but converges to a higher loss with constant learning rates, requiring a decaying learning rate to convergence zero training loss. In the interpolation regime, SGD and SVRG both achieve linear convergence, as known in [1,2].
>
> Q: I wonder if the phenomena still holds with much larger datasets.
> >> We will try the same experiments with the larger datasets like ImageNet. But at least 32 groups are needed to generate one line in our plots. With larger-scale experiments more time-consuming, we cannot guarantee all of them will be done timely before the end of rebuttal.
> [1]. Siyuan Ma, Raef Bassily, and Mikhail Belkin. The power of interpolation: Understanding the effectiveness of SGD in modern over-parametrized learning. In ICML, volume 80 of Proceedings of Machine Learning Research, pp. 3331–3340. PMLR, 2018.
> [2]. Mark Schmidt and Nicolas Le Roux. Fast convergence of stochastic gradient descent under a strong growth condition. arXiv preprint arXiv:1308.6370, 2013.

---

### Official Review · AnonReviewer2 · 2019-10-18
**Official Blind Review #2**

**Rating:** 6

**Review:**

This paper compares SGD and SVRG (as a representative variance reduced method) to explore tradeoffs. Although the computational complexity vs overall convergence performance tradeoff is well-known at this point, an interesting new perspective is the comparison in regions of interpolation (where SGD gradient variance will diminish on its own) and label noise (which propogates more seriously in SGD vs SVRG). The analysis is done on a simple linear  model with regression, with some experiments on simulations, MNIST, and CIFAR.

Overall, I find the paper insightful and the nice and neat breakdowns of the sources of noise nicely interpretable. A weakness is that the regression model and linear separation is a bit oversimplified, and may not really capture the subtleties in deeper models. However, I didn't find the conclusions particularly controversial, so it's not obvious that the model is wrong--just very simple.

How are step sizes chosen in the experiments? In general, a huge benefit of variance reduction is the ability to use constant step sizes. Can the authors elaborate on a comparison between SGD with decaying step size vs SVRG with constant step size?

One suggestion I would push for is to extend the experiments in the plots. In a lot of cases it doesn't really seem like the experiment is done running, e.g. fig 2 (b), 4 (b), and it's hard to make sweeping statements about the final loss without running to that point. Since many of the experiments seem to be on relatively small datasets and easier models, this should not be too burdensome.

While I like the breakdown of M vs m (for when the data is i.i.d.), I would say that the assumption that data is i.i.d. is not very realistic. That being said this is not a huge negative for this paper because both scenarios are considered.

minor stuff:
typo in theorem 4 (decay rate)

Post rebuttal: I read the comments and all the concerns are addressed. I don't really have any more major concerns about the paper.

**Experience Assessment:**

I have published one or two papers in this area.

**Review Assessment: Checking Correctness Of Derivations And Theory:**

I assessed the sensibility of the derivations and theory.

**Review Assessment: Checking Correctness Of Experiments:**

I assessed the sensibility of the experiments.

**Review Assessment: Thoroughness In Paper Reading:**

I read the paper at least twice and used my best judgement in assessing the paper.

---

> ### Author Response · Authors · 2019-11-15
> **Simplified linear models can still provide insightful intuitions to neural networks, shown also by our empirical results.**
>
> Thanks for your constructive feedback.
>
> Q: The regression model and linear separation is a bit oversimplified, and may not really capture the subtleties in deeper models.
> >> We do not claim our theory for regression model could directly translate to real deep nets, since directly analyzing the behavior of SGD and SVRG on neural nets which is generally hard (esp. for finite horizon analysis), but such analysis on linear model could possibly give rise to the intuitions of neural nets behind, because of the connection to kernel regression provided by NTK (see the second bullet point in the response to all reviewers). Such connections were also demonstrated in our underparameterized neural networks experiments.
>
> Q: How are step sizes chosen in the experiments? In general, a huge benefit of variance reduction is the ability to use constant step sizes. Can the authors elaborate on a comparison between SGD with decaying step size vs SVRG with constant step size?
> >> We conducted all of our experiments with constant step size and then plotted the minimum loss among all these hyperparameters of one computational budget. Specifically, for numerical simulations, 50 step sizes were chosen from 1.5 to 0.01. For the experiments on MNIST and CIFAR10, we picked 8 learning rates varying from 0.3 to 0.001.
>
> Q: In a lot of cases it doesn't really seem like the experiment is done running, e.g. fig 2 (b), 4 (b), and it's hard to make sweeping statements about the final loss without running to that point.
> >> Thanks for your suggestions. We extended the plots in Fig 4 (b) and updated the plot in our new version. The number of epoch is changed from 96 to 192 but the progress is limited with SVRG still not attaining global minimum.  As for the numerical experiment of Fig 2 (b), its y-axis is log scaled. We think it already runs to the optimal point when the loss attains $10^{-10} $.

---

### Official Review · AnonReviewer3 · 2019-10-22
**Official Blind Review #3**

**Rating:** 1

**Review:**


This paper aims to compare SGD and SVRG in deep learning, motivated by recent results that SGD performs better than SVRG, despite the latter's theoretical optimality.
The idea in the paper is to study this problem through linear regression by establishing
some asymptotic bounds for both SGD and SVRG. By looking into the terms of these bounds one can initiate a comparative study. A mixed picture is presented in the experiments which roughly agrees with some of the authors' claims.

There are, however, several important issues with the paper that require a major revision:

1) The connection between neural networks is never really established. There is also an obscure relationship between 'overparameterized/underparaterized' neural networks and 'without/with label noise'. While this relationship is important to switch our attention to a much simpler problem, the specifics are not explicated.

2) The theoretical content is not novel. All results on second moments (and more) are well known.
For example, [4] have both non asymptotic analysis, and a characterization of sampling variance for general SGD --- the assumptions of normal X with diagonal variance are very restricting (and unnecessary).
Additionally, the assumption of \theta_\star = 0 is not exactly WLOG.

3) The related work is not well cited. Examples:

 3a) "Instead of using the full gradients, the variants of SGD..."
  The citations for SGD here are a bit confusing: Robbins and Monro never talked about SGD; Duchi et al is not about standard SGD, and so on. Better references are [1, 2].

 3b) "The sampling variance and the slow convergence of SGD have been studied extensively
in the past (Robbins & Monro, 1951; Polyak & Juditsky, 1992; Bottou, 2010)."
None of this paper studies sampling variance of SGD. RM (1951) only study convergence of stochastic approximation. PJ (1992) is about iterate averaging. Bottou (2010) is also not about sampling variance, and only covers convergence on a high-level.
Look at [4] for the sampling variance of SGD procedures; also [5, 6].

3c) "Our main analysis tool is very closely related to recent
work studying the dynamics of gradient-based stochastic methods."
Misses important prior work in stochastic approximation dynamics.
Look at [7].


[1] Zhang, "Solving large scale linear prediction problems using gradient descent algorithms" (2004)
[2] Bottou, "Large-Scale Machine Learning with Stochastic Gradient Descent" (2010)
[3] Amari, "Natural gradient works efficiently in learning" (1998)
[4] Toulis and Airoldi, "Asymptotic and finite-sample properties of estimators
based on stochastic gradients" (2017)
[5] Li et al, "Statistical inference using SGD" (2017)
[6] Chen et al, "Statistical Inference for Model Parameters in Stochastic Gradient Descent" (2016)
[7] Kushner and Yin, " Stochastic approximation and recursive algorithms and applications" (2003)

**Experience Assessment:**

I have published in this field for several years.

**Review Assessment: Checking Correctness Of Derivations And Theory:**

I carefully checked the derivations and theory.

**Review Assessment: Checking Correctness Of Experiments:**

I assessed the sensibility of the experiments.

**Review Assessment: Thoroughness In Paper Reading:**

I read the paper thoroughly.

---

> ### Author Response · Authors · 2019-11-15
> **Theoretical contents are not known. Error bounds is not the same as exact expected loss at step t, which is necessary in our analysis. Please consider updating the score.**
>
> Thanks for your constructive feedback. Here are some responses to the concerns you raised.
>
> 1. The connection to neural networks:
>
> In section 1.1, we mentioned neural tangent kernel (NTK) to connect our theoretical analysis with/withour label noise to the experiments in underparametrized/overparametrized regime. In fact, for over-parametrized neural networks, there are a bunch of work that draws connections between neural networks and linear regression model [2-5]. To be precise, when the number of parameters $p$ greatly exceeds the number of data $n$, it can be shown by [5] that the parameter $\theta$ moves only a small amount w.r.t. some initialization ${\theta}_0$, and hence it is possible to linearize the model around $\theta_0$, i.e. $f(x;{\theta}) = \nabla_{{\theta}} f(x; {\theta}_0)^\top {\theta}$, for ${\theta} = {\theta}-{\theta}_0$ the distance parameters moved during training. This is exactly the linear regression model, and this notion has already been adopted in [4, Section 1].
>
> The main difference between over- and under-parametrized neural nets is the ability for the function space to cover the target function, i.e. the so-called “interpolation regime”. For under-parametrized neural nets, this model is related to the linear regression model with label noise. We do not directly analyze the behavior of SGD and SVRG on neural nets which is generally hard (esp. for finite horizon analysis), but such analysis on linear model could possibly give rise to the intuitions of neural nets behind.
>
> 2. The theoretical content novelty.
>
> We would like to emphasize that our paper derives the exact expected loss at t-step for the noisy least square model for both SGD and SVRG methods, instead of an non-asymptotic upper bound as given in [4]. Deriving the exact loss is necessary because we need to compare the two methods’s performance at each time step and upper bound cannot provide any valid comparisons. The dynamics we derived (Eq.5 and Lemma 2) are then used to run the numerical simulations, and compare the two methods in Section 4.1.
> In addition, [6] derives non-asymtotic upper bounds for implicit SGD methods, which does not contain SVRG. Our main contribution lies in the analysis of SVRG to explain its dilemma when applied in deep learning tasks. Hence we believe our theoretical results for the second moment of SVRG are novel.
>
> 3. Related work.
> We have revised the related work and change them properly based on your suggestions.
>
> [1] Arthur Jacot, Franck Gabriel, and Clement Hongler. Neural tangent kernel: Convergence and generalization in neural networks. Advances in Neural Information Processing Systems, 31, 2018.
> [2] Allen-Zhu, Zeyuan, Yuanzhi Li, and Zhao Song. A Convergence Theory for Deep Learning via Over-Parameterization. International Conference on Machine Learning. 2019.
> [3] Du, Simon, Jason Lee, Haochuan Li, Liwei Wang, and Xiyu Zhai. Gradient Descent Finds Global Minima of Deep Neural Networks. International Conference on Machine Learning. 2019.
> [4] Hastie Trevor, Andrea Montanari, Saharon Rosset, and Ryan J. Tibshirani. Surprises in high-dimensional ridgeless least squares interpolation. arXiv preprint arXiv:1903.08560 2019.
> [5] Chizat, Lenaic, Edouard Oyallon, and Francis Bach. On Lazy Training in Differentiable Programming. arXiv preprint arXiv: 1812.07956 2018.
> [6] Toulis and Airoldi, "Asymptotic and finite-sample properties of estimators based on stochastic gradients" (2017)

---

### Public Comment · ~Sebastian_U_Stich1 · 2019-10-15
**SCDG vs SGD?**

Hi,
I just read your paper.

In addition to (Sebbouh et al., 2019) you might also find the (same) method in https://arxiv.org/abs/1805.00982 of relevance.

It would be very interesting to see a comparison of SGD and SVRG in deep learning settings. Did you do such comparisons?

Also, methods like SCDG (https://arxiv.org/abs/1609.03261) seem to be a bit cheaper than SVRG; do you know if they perform better (or more efficiently) than SVRG?

Thanks,

---

> ### Author Response · Authors · 2019-10-23
> **Comparison in deep learning settings.**
>
> Thanks for your interest in our paper. We will add k-SVRG to the related work.
>
> For experiments with neural nets (deep learning), in section 4, we compared SVRG and SGD in both the underparameterized and the overparameterized regimes, using MLP for MNIST and CNN for CIFAR-10. In the overparameterized setting, the MLP for MNIST contains two hidden layers, each layer having 1024 neurons; the CNN for CIFAR-10 has one 64-channel convolutional layer, one 128-channel convolutional layer followed by one 3200 to 1000 fully connected layer and one 1000 to 10 fully connected layer. As for the underparameterized models, you can check section 4.2 for more details about our network architecture. The results from both regimes match our theoretical analysis. Underparameterized neural networks match our theoretical results with label noise and overparameterized models’ performance match the theoretical results without label noise.
>
> With regards to SCSG, the method replaces the full-batch gradient with the gradient of a medium-size batch as a cheaper way to reduce the variance. Similarly, our analysis also assumes that SVRG uses a batch gradient to reduce the variance, instead of the full-batch gradient. In our numerical simulation and empirical studies, we tried to find the optimal batch size and snapshot interval for SVRG under a fixed computational budget, compared to SGD. Hence we believe the results we obtained should also apply to SCSG.

---

### Author Response · Authors · 2019-11-15
**New version updated. Common concerns among reviewers are addressed below.**

Thanks for constructive feedback from all reviewers. We have found and fixed a mistake in Appendix B. And all simulations were rerun based on new theorems, and our previous conclusions still hold. In the new theorems, the coefficient matrix in Eq.6 of Theorem 2 changes from a diagonal matrix to a positive definite symmetric matrix. In this case, the second moment of each parameter no longer evolve independently. Besides, we have revised all typos pointed out by reviewer 1 and reviewer 2.  Below are some crucial points we need to clarify about our paper.

1. Exact loss at step $t$ vs. error bound:
We would like to emphasize that our paper derives the exact expected loss at t-step for the noisy least square model for both SGD and SVRG methods, instead of an non-asymptotic upper bound as given in citations referred by reviewer 3 [1]. Deriving the exact loss is necessary because we need to compare the two methods’ performances at each time step and upper bound cannot provide any valid comparisons.

2. Connection of overparametrized/underparametrized to without/with label noise:
In section 1.1, we mentioned neural tangent kernel (NTK) to connect our theoretical analysis with/withour label noise to the experiments in underparametrized/overparametrized regime. In fact, for over-parametrized neural networks, there are a bunch of work that draws connections between neural networks and linear regression model [2-5]. To be precise, when the number of parameters $p$ greatly exceeds the number of data $n$, it can be shown by [5] that the parameter $\theta$ moves only a small amount w.r.t. some initialization ${\theta}_0$, and hence it is possible to linearize the model around $\theta_0$, i.e. $f(x;{\theta}) = \nabla_{{\theta}} f(x; {\theta}_0)^\top {\theta}$, for ${\theta} = {\theta}-{\theta}_0$ the distance parameters moved during training. This is exactly the linear regression model, and this notion has already been adopted in [4, Section 1].

The main difference between over- and under-parametrized neural nets is the ability for the function space to cover the target function, i.e. the so-called “interpolation regime”. For under-parametrized neural nets, this model is related to the linear regression model with label noise. We do not directly analyze the behavior of SGD and SVRG on neural nets which is generally hard (esp. for finite horizon analysis), but such analysis on linear model could possibly give rise to the intuitions of neural nets behind.

3. Convergence rate of SVRG vs. SGD:
Our paper presents results that may seem to contradict the well-known fact that SVRG converges faster than SGD, and this creates some confusion to the reviewers. We would like to clarify this paradox. First of all, we would like to emphasize that we derived the exact expected loss at t-step for both algorithms under the noisy linear square model, instead of any asymptotic convergence rate results. We then compared the two algorithms numerically by running the dynamics with constant learning rates (Section 4).
In the traditional analysis, one needs a decaying learning rate schedule for SGD so that it can converge. SVRG on the other hand does not require a decay learning rate hence achieving a faster convergence rate. In contrast to the standard analysis, we used fixed learning rate schedule for both algorithms. In the case with label noise (under-parameterized), we observed that SGD achieved lower loss that SVRG for the first part of the training, but SGD converged to a higher loss than SVRG, as expected. In the case without label noise (interpolation regime), our experimental results agreed with what’s known in prior work [6,7]: SGD and SVRG both achieve linear convergence.

[1] Toulis and Airoldi. Asymptotic and finite-sample properties of estimators based on stochastic gradients. (2017)
[2] Arthur Jacot, Franck Gabriel, and Clement Hongler. Neural tangent kernel: Convergence and generalization in neural networks. Advances in Neural Information Processing Systems, 31, 2018.
[3] Allen-Zhu, Zeyuan, Yuanzhi Li, and Zhao Song. A Convergence Theory for Deep Learning via Over-Parameterization. International Conference on Machine Learning. 2019.
[4] Hastie Trevor, Andrea Montanari, Saharon Rosset, and Ryan J. Tibshirani. Surprises in high-dimensional ridgeless least squares interpolation. arXiv preprint arXiv:1903.08560 2019.
[5] Chizat, Lenaic, Edouard Oyallon, and Francis Bach. On Lazy Training in Differentiable Programming. arXiv preprint arXiv: 1812.07956 2018.
[6]. Siyuan Ma, Raef Bassily, and Mikhail Belkin. The power of interpolation: Understanding the effectiveness of SGD in modern over-parametrized learning. In ICML, volume 80 of Proceedings of Machine Learning Research, pp. 3331–3340. PMLR, 2018.
[7]. Mark Schmidt and Nicolas Le Roux. Fast convergence of stochastic gradient descent under a strong growth condition. arXiv preprint arXiv:1308.6370, 2013.

---

### Decision · Program_Chairs · 2019-12-19

**Decision:**

Reject

**Comment:**

Two reviewers as well as the AC are confused by the paper—perhaps because the readability of it should be improved?  It is clear that the page limitation of conferences are problematic, with 7 pages of appendix (not part of the review) the authors may consider another venue to publish.  In its current form, the usefulness for the ICLR community seems limited.